# SARS-CoV-2 Neutralizing Responses in Various Populations, at the Time of SARS-CoV-2 Variant Virus Emergence: Evaluation of Two Surrogate Neutralization Assays in Front of Whole Virus Neutralization Test

**DOI:** 10.3390/life12122064

**Published:** 2022-12-09

**Authors:** Stephane Marot, Djeneba Bocar Fofana, Philippe Flandre, Isabelle Malet, Karen Zafilaza, Valentin Leducq, Diane Vivien, Sarah Mrabet, Corentin Poignon, Vincent Calvez, Laurence Morand-Joubert, Anne-Geneviève Marcelin, Joel Gozlan

**Affiliations:** 1Sorbonne Université, INSERM, Institut Pierre Louis d’Epidémiologie et de Santé Publique (iPLESP), F-75012 Paris, France; 2Department of Virology, Assistance Publique-Hôpitaux de Paris (AP-HP), Pitié Salpêtrière Hospital, F-75013 Paris, France; 3Department of Virology, Assistance Publique-Hôpitaux de Paris (AP-HP), Saint-Antoine Hospital, F-75012 Paris, France; 4Centre de Recherche Saint-Antoine, “Cancer Biology and Therapeutics”, Sorbonne Université & INSERM UMR_S 938, F-75012 Paris, France

**Keywords:** SARS-CoV-2, COVID-19, neutralizing antibodies, neutralization surrogate assays, SARS-CoV-2 variants

## Abstract

The SARS-CoV-2 neutralizing antibodies response is the best indicator of effective protection after infection and/or vaccination, but its evaluation requires tedious cell-based experiments using an infectious virus. We analyzed, in 105 patients with various histories of SARS-CoV-2 infection and/or vaccination, the neutralizing response using a virus neutralization test (VNT) against B.1, Alpha, Beta and Omicron variants, and compared the results with two surrogate assays based on antibody-mediated blockage of the ACE2-RBD interaction (Lateral Flow Boditech and ELISA Genscript). The strongest response was observed for recovered COVID-19 patients receiving one vaccine dose. Naïve patients receiving 2 doses of mRNA vaccine also demonstrate high neutralization titers against B.1, Alpha and Beta variants, but only 34.3% displayed a neutralization activity against the Omicron variant. On the other hand, non-infected patients with half vaccination schedules displayed a weak and inconstant activity against all isolates. Non-vaccinated COVID-19 patients kept a neutralizing activity against B.1 and Alpha up to 12 months after recovery but a decreased activity against Beta and Omicron. Both surrogate assays displayed a good correlation with the VNT. However, an adaptation of the cut-off positivity was necessary, especially for the most resistant Beta and Omicron variants. We validated two simple and reliable surrogate neutralization assays, which may favorably replace cell-based methods, allowing functional analysis on a larger scale.

## 1. Introduction

The magnitude of neutralizing antibody (NAb) responses against the SARS-CoV-2 virus is a key factor for the outcome of coronavirus diseases in 2019 (COVID-19) [1,2] and long-term protection after infection [3] or vaccination [4]. However, quantitation of anti-spike-receptor binding domain (RBD) antibodies (Abs) varies in their accuracy to predict an effective neutralizing activity for the SARS-CoV-2 virus [5,6], which relies on many factors, such as antigen used to capture Abs, the commercial test used, the origin of Abs (vaccine vs infection) and individual’s characteristics 

Another issue is the worldwide emergence of SARS-CoV-2 variants that escape the neutralizing action of Abs due to deletions and substitutions in functional regions of RBD. The Alpha variant of concern (VOC) (lineage B.1.1.7) demonstrates a 3-fold reduction of NAb titers in mRNA-vaccinated people, compared to the historical B.1 virus [7,8,9]. The Beta VOC (lineage B.1.351) shows a stronger neutralizing escape, with up to a 14-fold decrease in NAb titers, in vaccinated or convalescent individuals [7,9] as well as the Omicron VOC (lineage B.1.1.529), with the higher neutralization escape evaluated at 18- to 53-fold decrease in NAbs titers from vaccinated or convalescent individuals [10].

The reference method to assess the neutralizing activity of sera is the virus neutralization test (VNT), which uses infectious particles in cell cultures. These assays rely on the in vitro infection of ACE-2-expressing cells with natural or modified (GFP or luciferase-expressing) infectious viruses or pseudotypes expressing SARS-CoV-2 spike protein in the presence of serial dilutions of sera, followed by the monitoring of virus gene expression. These assays are time-consuming, potentially biohazardous and require biosafety level three facilities, which impede their routine use for large studies. The development of reliable surrogate tests is therefore of paramount importance to evaluate functional responses to infection and/or vaccination at a large scale. 

We analyzed in various clinical situations the neutralizing activity of sera, evaluated by the reference VNT and compared VNT results with the following two commercial surrogate tests based on antibody-mediated blockage of ACE2-RBD interaction: a lateral flow assay (ichroma™ COVID-19 nAb, Boditech, Chuncheon, South Korea) and an ELISA (SARS-CoV-2 Surrogate Virus Neutralization Test, Genscript, Piscataway, NJ, USA). 

We performed VNT using the B.1 historical strain circulating in France before fall 2020, as well as two main variants (Alpha and Beta) circulating during 2021 and the Omicron variant (B.1.1.529) circulating from the end of 2021. 

Viral neutralization was also compared with the quantification of anti-RBD antibodies by high-input enzyme-linked chemiluminescent assay in order to define serological cut-off indicating significant neutralization activities of sera.

## 2. Materials and Methods

### 2.1. Populations Tested

We analyzed 105 patients, classified into five groups (Table 1).

Group 1 “Naive + 2D”: non-infected individuals, vaccinated with two doses of mRNA vaccine BNT162b2 (*n* = 35). Among them, 25 were immuno-competent, whereas 10 were immunosuppressed, suffering from hematological disease. Sera were collected one to three weeks after the second injection. 

Group 2 “Naive + 1D”: non-infected immunocompetent individuals, analyzed after one dose of SARS-CoV-2 vaccine (half vaccination schedule) (*n* = 14). Sera were collected three to four weeks after injection. 

Group 3 “COVID-19 + 1D”: Recovered COVID-19 patients, vaccinated with one vaccine dose: (*n* = 24, BNT162b *n* = 9; ChAdOx1 AZ, *n* = 15). Vaccination was performed between 4 and 12 months after infection, and sera were collected one to four weeks after injection. 

Group 4 “COVID-19”: non-vaccinated patients recovered from COVID-19: *n* = 22. Sera were collected from 4 to 12 months after infection. 

Group 5 “HCoV”: Patients recovered from other human coronavirus (HCoV) infections dating from pre-pandemic SARS-CoV-2 period (*n* = 10).

Other sera (*n* = 111), collected before July 2019 (pre-pandemic period), were analyzed by the surrogate neutralization assays to complete specificity analysis. Sixty were collected during other acute infections whereas 51 were randomly chosen. 

This retrospective study was carried out in accordance with the Declaration of Helsinki without addition to standard of care procedures. The study was approved by the Institutional Review Board n° IRB00011642 (Comité d’Ethique de Recherche en Maladies Infectieuses Tropicales) under the N° CER-MIT 2022-0504. The data collection has been declared to Sorbonne Université under the number 2020-025. Written informed consent for participation in this study was obtained from all participants.

### 2.2. Virus Neutralization Test 

The neutralizing activity of sera was assessed with a whole virus replication assay as previously described [11], using the following four SARS-CoV-2 isolates: the B.1, B.1.1.7, B.1.351 and B.1.1.529 strains (GenBank accession number MW322968, MW633280, MW580244 and GISAID accession ID 11458826). A microscopy examination was performed on day 4 to assess the cytopathic effect (CPE). Nab titers are the highest serum dilution displaying 100% (NT100) inhibition of the CPE. The same positive control serum was added to each experiment to assess the repeatability. A titer above 5 was considered positive. 

### 2.3. Neutralizing Antibodies Surrogate Lateral Flow Assay

A new SARS-CoV-2 surrogate neutralization assay, based on antibody-mediated blockage of ACE-2-Spike protein interaction was used, according to manufacturer’s recommendations (ichroma™ COVID-19 nAb, Boditech, South Korea). Briefly, sera are pre-incubated with a fluorescence-labeled SARS-CoV-2 RBD antigen, in a detection buffer containing ACE-2-biotin conjugate. The mixture is loaded in a lateral flow nitrocellulose matrix, where covalent complexes RBD-ACE-2-biotin are immobilized on the streptavidin capture “Test line”. The more Nabs are present, the more it interferes with the binding of labeled RBD to ACE-2-biotin, which results in less fluorescence. According to fabricant’s instructions, a fluorescence inhibition above 30% is considered positive. 

### 2.4. Neutralizing Antibodies Surrogate ELISA Assay

This semi-quantitative ELISA assay (SARS-CoV-2 Surrogate Virus Neutralization Test, Genscript, USA), which is also based on antibody-mediated blockage of ACE-2-Spike protein interaction, has been described previously [12]. Briefly, sera are pre-incubated with horseradish peroxidase (HRP)-labeled SARS-CoV-2 RBD, then added to ACE-2 coated ELISA plates. RBD-ACE2 interaction is biochemically quantified by adding a substrate TMB solution. The more Nabs are present, the more they inhibit the binding of labeled RBD to ACE-2, which results in less signal. According to the fabricant, an inhibition above 30% is considered positive. 

### 2.5. Anti-Spike Antibodies EIA Quantitation

Quantification of anti-spike RBD IgG antibodies was assessed by high-input chemiluminescence assay, the SARS-CoV-2 IgG II Quant (Abbott, Rungis, France) on Alinity i platform, according to manufacturer’s instructions. The cut-off positivity is at 7.1 binding antibody units per milliliter (BAU/mL). 

### 2.6. Statistical Analysis

Quantitative variables are described by median and Interquartile Range (IQR), while categorical variables are described in percent. Between-group comparisons were carried out using Wilcoxon nonparametric test. Spearman correlations were computed between several continuous variables. Receiver operating characteristic (ROC) curve analysis and Youden index were used to identify optimized thresholds (cut-off indices). Analyses were performed using R (package ROCit, https://www.R-project.org/ accessed on 19 October 2022).

## 3. Results

### 3.1. Anti-RBD IgG Antibody Levels and nAb Titers Assessed by the VNT Assays

Figure 1 summarizes, for each group, nAb titers for the four strains assessed (Figure 1A,C) and anti-RBD levels quantified by EIA (Figure 1B). Complete data are in the Appendix A.

Sera from the group with half vaccination schedule (Naive 1D) only displayed weak—and inconstant—neutralization activity against B.1 and the Alpha variant and almost no activity for the Beta variant (neutralization activity was not assessed against the Omicron variant), despite moderate but detectable IgG anti-RBD antibodies observed in all patients (from 7 to 173 BAU/mL). On the contrary, almost all patients receiving 2 doses of mRNA vaccine (Naive 2D) displayed significant neutralization activity against the B.1, Alpha and Beta variants, but only 31.4% (*n* = 11/35) showed neutralization activity against the Omicron variant (Figure 1A). Immunosuppressed patients had neutralization activity that appeared reduced and inconstant for the Beta variant and almost absent for the Omicron variant (Figure 1C). 

Importantly, all sera from recovered COVID-19 patients kept a significant neutralization activity against B.1 and the Alpha variant up to 12 months after recovery. However, we observed no significant neutralization against Beta and Omicron variants for 45.5% (*n* = 10/22) and for 77.3% (*n* = 17/22), respectively.

Finally, the best responses (*p* < 0.001 for Nab titers and *p* = 0.04 for anti-RBD titers, compared with “Naive + 2D”) were observed for convalescent COVID-19 patients vaccinated with one dose of vaccine, where all patients harbored a significant neutralization titer for the four variants and high levels of anti-RBD Abs.

### 3.2. Specificity of the Surrogate Assays

We analyzed 121 sera from the pre-pandemic period to assess specificity. A weak positivity (from 31 to 54% inhibition) was observed from six samples with the Boditech assay, for a specificity of 95% (IC95: 91.1–98.9%). Five of these samples were collected during other acute infections (2 primary EBV, 2 acute HAV, and 1 acute HEV infection), and the last was from a chronically HCV-infected patient. The specificity of the Genscript ELISA test was even better since only one sample displayed a weak positive result (35% inhibition), for a specificity of 99.2% (IC95: 97.6–100%). This serum was collected after an acute influenza A infection. The complete results of specificity analysis are detailed in Appendix A.

### 3.3. Correlations and Performances of the Surrogate Assays Compared to the VNT

Figure 2 summarizes the correlations between the surrogate Boditech (Figure 2A) or Genscript (Figure 2B) assays and VNT for B.1, Alpha, Beta and Omicron variants. The surrogate markers showed high correlations with NAb titers for B.1, Alpha and Beta variants, with Spearman correlation indices between 0.89 and 0.94 for the Boditech assay and between 0.84 and 0.86 for the Genscript assay. The correlation was lower for the Omicron variant, with a decrease of the Spearman correlation index to 0.80 for the Boditech assay and 0.66 for the Genscript assay.

We performed ROCs analyses (Figure 3) to evaluate the performances of the surrogate tests in front of the reference VNT and determine the optimal cut-off to be used with these surrogate assays, for each variant assessed.

For the lateral flow assay (Figure 3A), area under curves (AUC) were excellent for the four variants (0.96, 0.96, 0.97 and 0.94, respectively). The optimal cut-off determined for this assay for the B.1 strain (25.9%) was in accordance with the 30% inhibition cut-off suggested by the fabricant. For the more resistant Alpha, Beta and Omicron variants, the good AUC necessitates raising this cut-off to 47.5% for Alpha and Beta and to 87.6% for Omicron variants. 

For the ELISA assay (Figure 3B), the AUCs were excellent for the three more sensitive B.1, Alpha and Beta variants (AUC = 0.94, 0.95 and 0.97) and slightly decreased to 0.85 for the more resistant Omicron strain. Again, these good concordances were only observed with the increase of cut-off positivity to respectively 76.1, 85.7, 90.6% and 99.5% inhibition, which is higher than the 30% cut-off suggested by the fabricant.

### 3.4. Correlations between VNT and Quantitation of Anti-RBD Antibodies

We then analyzed the correlation between the quantification of anti-RBD and NAb titers and determined if a level of anti-RBD may predict a significant neutralizing activity of the serum (NAb titer ≥ 10). A significant correlation between anti-RBD levels and NAb titers was observed for the four variants tested in VNT, which were strong for B.1, Alpha, Beta and moderate for the Omicron variant (Spearman’s rank at 0.90, 0.86, 0.85 and 0.64, respectively), with the lowest slope observed for this most resistant Omicron isolate (Figure 4).

Using two-sided generalized maximally selected statistics analysis, we determined the optimal cut-off for anti-RBD levels able to predict neutralizing activity against the four variants (Table 2). These cut-off levels were 173, 173, 732 and 1886 BAU/mL for B.1, Alpha, Beta and Omicron variants.

## 4. Discussion

Accurate determination of effective immune responses conferred by SARS-CoV-2 infection and/or vaccination is a key issue to anticipate individual protection and adapt global vaccination strategies, in front of the worldwide emergence of potential escape variants, individual variability of immune response and waning of immune response with time [13].

Neutralizing antibodies are a major effector of this response, representing a valuable marker for immune protection. Most of these antibodies inhibit the binding of the viral spike antigen to its cellular receptor ACE-2, allowing the development of surrogate assays mimicking this virus-cell interaction and measuring its inhibition by antibodies. These surrogate markers may favorably replace tedious assays assessing neutralizing titers in cell-based methods, thus allowing their use in a larger scale perspective. 

We first analyzed the neutralization response in different clinical groups, using the reference VNT method for the four variants circulating in France during the period of analysis. We confirmed the good response conferred by a full vaccination for B.1, Alpha and Beta variants, whereas vaccine-induced responses were infrequent (31.4%) for the Omicron variant [10,14,15]. A single vaccine dose, administered to patients recovering from an ancient COVID-19, was effective for the four variants tested, as previously reported [10,16,17]. On the contrary, an incomplete vaccination only induced a weak and inconstant response, especially for the escape mutant Beta, for which neutralizing activity was almost null. In accordance with a recent work [18], we also demonstrated that naturally secreted antibodies keep for several months an effective neutralizing activity for B1 and Alpha strains, but this activity is lost for half of the recovered patients for the B.1.351 stain and almost null for the Omicron variant [10]. 

When we looked at whether anti-RBD IgG levels may predict a significant neutralization activity, we were able to determine various cut-off levels according to the virus tested in VNT. Importantly, these “functional” levels were above the cut-off proposed by the fabricant for positivity (7.1 BAU/mL) and rose to very high levels (1886 BAU/mL) for the Omicron strain, circulating worldwide. Moreover, there is an individual heterogeneity in the functional activity of anti-RBD, especially for this dominant Omicron strain (Table 2), making the functional assay mandatory.

We therefore evaluated two surrogate assays for quantifying Abs-mediated blockage of RBD-ACE2-protein interactions. The specificity of both assays appeared excellent, with infrequent weak reactivity observed for pre-pandemic samples collected during other viral infections. 

These assays showed a high correlation with VNT, with the AUC of the ROC analysis ranging from 0.85 to 0.97. Despite the fact that recombinant RBD proteins used in these assays were produced before the worldwide emergence of resistant VOCs, their performances were maintained for two major Alpha and Beta VOCs circulating in France at the beginning of 2021 and also for the most resistant Omicron strain that has been predominant since then. However, these good performances require an adaptation of the positivity cut-off, especially for the most resistant Omicron stain and for the ELISA assay. In front of the excellent specificity of these 2 assays, we believe that this cut-off adaptation (which is suggested in the package inserts) is due to an excess of sensitivity of these techniques rather than a lack of specificity. A recent paper evaluating the Boditech Lateral Flow assay also showed its ability to accurately predict an effective neutralization in front of various SARS-CoV-2 variants, at the condition to adapt the cut-offs to each VOCs [19].

Besides their good performances, the major advantage of these surrogate assays is their ease, allowing their use in any laboratory without the need for biosafety facilities. Another advantage is their rapidity (90 min for the 96-well ELISA plate and 20 min for the lateral flow assay). The ELISA test can be automatized in a microplate manager automate, whereas the lateral flow assay requires a simple manual procedure and a fluorescence reader. The individual format of the latter is a disadvantage, but several tests can be run in a row, which allows testing about 15 samples per hour. 

Other surrogate neutralization assays, also based on inhibition of ACE-2-Spike interactions, have been described and/or evaluated [20,21,22,23]. These assays displayed various performances in smaller cohorts than ours. A last surrogate assay has been improved by the incorporation of multiplexed trimeric SARS-CoV-2 spike proteins of several lineages, allowing simultaneous analysis of multiple variants in the same experiment [24]. This assay, which harbored high levels of sensitivity and specificity, is promising but not yet commercialized. 

These surrogate assays are attractive, but they do not evaluate the totality of the complex process driving SARS-CoV-2 infection of target cells. Proteolytic processing of S protein, as well as palmitoylation in its cysteine-rich domain (CRD), also contributes to virus-cell fusion and SARS-CoV-2 entry in the cell [25]. Antibodies that interfere with these modified domains and ACE2 may be neutralizing as well and are not evaluated in the surrogate assays. However, blockage of RBD and ACE2 is considered the major neutralizing mechanism, making these assays relevant. 

Our study had several limitations. First, cellular responses to SARS-CoV-2 are not evaluated by the assays we used. However, even if cell responses contribute to anti-SARS-CoV-2 immunity [26,27], neutralizing antibodies likely play the primary role in this process [1,2,3,4] and represent the easiest marker to assess protection. Second, we did not include in our evaluation the Delta variant since it was not present in France when we started the experiments and did not circulate anymore when we completed this study on the Omicron variant. Recent data indicate a reduced sensitivity of this variant to antibody-mediated neutralization [7,15]. However, this resistance appears modest (from 2 to 4-fold), which is less than the resistance of the Beta and Omicron strains evaluated in this study and should not significantly impact the mRNA-vaccine efficacy. 

In conclusion, we validated in this study two simple surrogate assays for antibody-mediated neutralization, in front of the reference method based on cell culture with living viruses. These surrogate markers appeared to be reliable, even for the most resistant Omicron variant, at the condition needed to raise their positivity cut-off. Their potential usefulness is numerous, either from a clinical use or public health perspective. First, they can provide information on the effectiveness and durability of the immune response after infection, vaccination or passive immunization using monoclonal antibodies. They can give insights on the necessity –or not- to boost vaccination schedule in certain patients (elderly, transplanted or immunosuppressed) or at distance from the last injection. On the contrary, they may be used to save vaccine doses for individuals that are still protected, which can be a major issue in countries where sufficient vaccine supplies are still lacking.

## Figures and Tables

**Figure 1 life-12-02064-f001:**
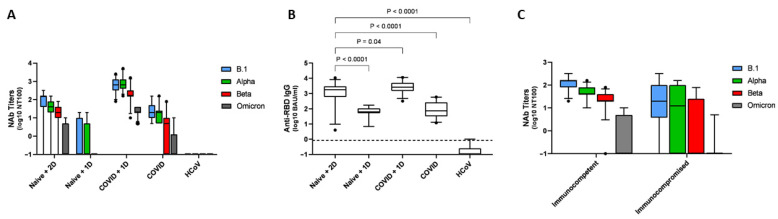
Neutralizing antibody (nAb) titer with 100% inhibition (NT100) against clinical strains of B.1, Alpha, Beta and Omicron variants and anti-RBD IgG levels of 105 participants (81 participants were tested against Omicron strain). (**A**) NT100 against the four variants by clinical group (Naïve + 2D: SARS-CoV-2 naïve participants with 2 vaccine injections; Naïve + 1D: SARS-CoV-2 naïve participants with 1 vaccine injection; COVID + 1D: previously infected SARS-CoV-2 participants with 1 vaccine injection; COVID: previously infected SARS-CoV-2 participants and hCoV: convalescent sera from previously infected participants with other human coronaviruses. (**B**) Anti-RBD IgG levels of participants by clinical group (**C**) NT100 against the four variants of SARS-CoV-2 naïve participants with 2 vaccine injections divided by their immunological status. Black horizontal lines indicate median NT100. Whiskers indicate 95% confidence interval. Black dot are outliers. Two-tailed P values were determined using the Mann–Whitney test and are reported on each panel.

**Figure 2 life-12-02064-f002:**
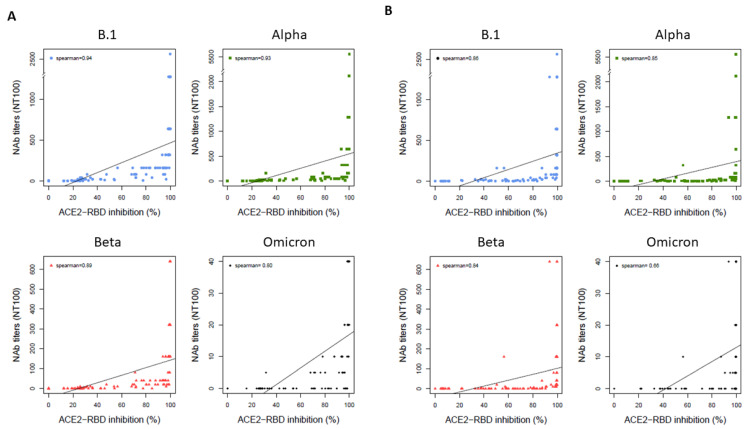
Correlations between the surrogate assays and neutralizing antibody (NAb) titer with 100% inhibition (NT100) for B.1, Alpha, Beta and Omicron variants. (**A**) Correlations for the surrogate lateral flow assay (Boditech) (**B**) Correlations for the surrogate inhibition ELISA assay (Genscript). Y-axis was variable for SARS-CoV-2 strains used.

**Figure 3 life-12-02064-f003:**
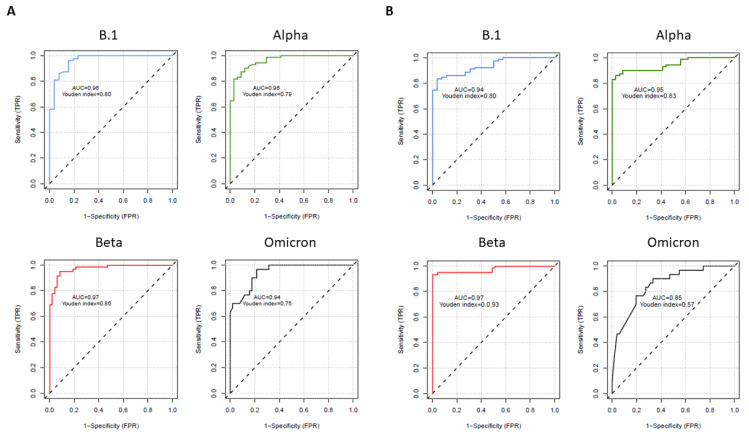
Receiver operating characteristic (ROC) curves for the surrogate assays against each SARS-CoV-2 variants tested (**A**) lateral flow assay (Boditech) (**B**) Inhibition ELISA assay (Genscript).

**Figure 4 life-12-02064-f004:**
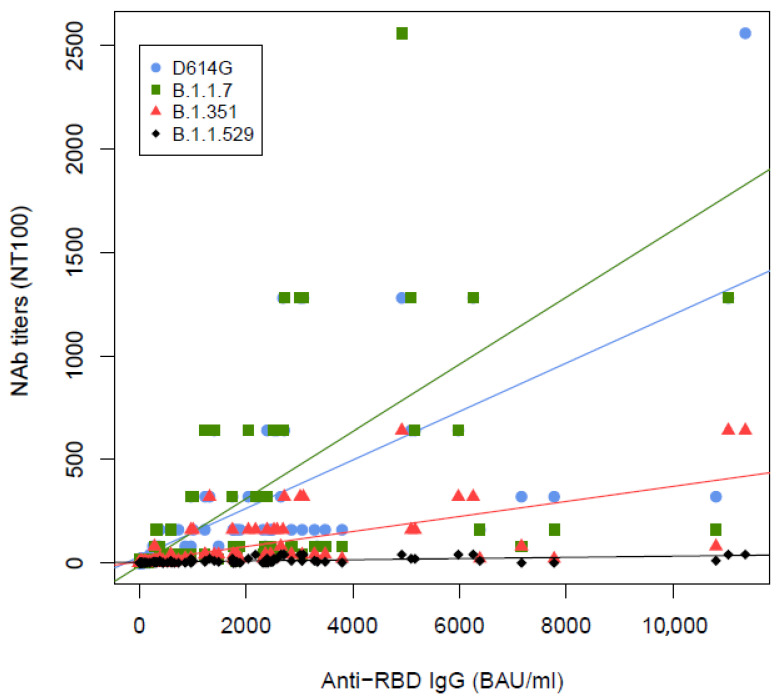
Correlations between quantification of anti-RBD IgG and Neutralizing antibody (NAb) titer with 100% inhibition (NT100) against clinical strains of B.1, Alpha, Beta and Omicron variants and anti-RBD IgG levels of 105 participants (81 participants were tested against Omicron strain).

**Table 1 life-12-02064-t001:** Characteristics of the populations.

Global Population Characteristics	*n* = 105
Male, *n* (%)	46 (43.8%)
Age (years), median [IQR]	50 [37–60]
**Subpopulation characteristics**	
**Naive individuals vaccinated with two doses of vaccine: “Naive + 2D”**	*n* = 35
Male, *n* (%)	10 (28.6%)
Age (years), median [IQR]	58 [50–64.5]
Time between 2nd vaccine dose and serum sampling (days), median [IQR]	7 [7–13.75]
**Naive individuals vaccinated with 1 dose of vaccine: “Naive + 1D”**	*n* = 14
Male, *n* (%)	4 (28.6%)
Age (years), median [IQR]	32.5 [29.5–34.75]
Time between the vaccine dose and serum sampling (days), median [IQR]	28 [27.25–28]
**COVID-19 patients vaccinated with 1 dose of vaccine: “COVID-19 + 1D”**	*n* = 24
Male, *n* (%)	16 (66.7%)
Age (years), median [IQR]	44.5 [32.75–51.25]
Time between the vaccine dose and serum sampling (days), median [IQR]	27 [20–28]
**Individuals recovered from COVID-19: “COVID-19”**	*n* = 22
Male, *n* (%)	9 (40.9%)
Age (years), median [IQR]	47.5 [39.5–55.5]
Time between symptom’s onset and serum sampling (days), median [IQR]	187 [182.5–238]
**Patients recovered from other human coronavirus infection: “HCoV”**	*n* = 10
Male, *n* (%)	7 (70%)
Age (years), median [IQR]	57.5 [43–64]
Time between diagnostic and serum sampling (days), median [IQR]	24.5 [16.75–32]

**Table 2 life-12-02064-t002:** Optimal thresholds values for SARS-CoV-2 anti-RBD IgG levels, for prediction of significant in vitro neutralization in front of the four variants evaluated.

SARS-CoV-2 Strain	Optimal Cut-Off for Anti-RBD IgG (BAU/mL)	Nab Titer < 1:10 *n* (%)	Nab Titer ≥ 1:10 *n* (%)	Number of Patients
B.1	<173.3	33 (79%)	9 (21%)	42
	≥173.3	1 (2%)	62 (98%)	63
Alpha	<173.3	36 (86%)	6 (14%)	42
	≥173.3	4 (6%)	59 (94%)	63
Beta	<731.7	53 (93%)	4 (7%)	57
	≥731.7	2 (4%)	46 (96%)	48
Omicron	<1886.4	45 (94%)	3 (6%)	48
	≥1886.4	16 (48%)	17 (52%)	33

## Data Availability

The data presented in this study are available on request from the corresponding author.

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
