# Peer review of "SARS-CoV-2 Neutralizing Responses in Various Populations, at the Time of SARS-CoV-2 Variant Virus Emergence: Evaluation of Two Surrogate Neutralization Assays in Front of Whole Virus Neutralization Test"

_life, 2022, doi:10.3390/life12122064_

Round 1
Reviewer 1 Report
The authors compare the SARS-CoV-2 neutralizing antibodies response by virus neutralization test and two comercially available surrogate neutralization assays. Their comparison involved also several SARS-CoV-2 variants. Overall, this is an interesting and valuable work showing that used surrogate assays displayed good correlation with the VNT.
The manuscript is well written, methods are presented clearly and in details, experiments and analyses were performed correctly. Also, the claims are supported by adequate references.
Reviewer 2 Report
The manuscript described the comparison of two surrogate assays with general SARS-CoV-2 detection methods. It is useful at the time of SARS-CoV-2 variant virus emergence.
However, there are some shortcomings.
1) Simple Summary and Abstract are not clear enough to represent the title and the whole manuscript.
2) No description for two surrogate assays of Lateral Flow Boditech and ELISA Genscript in “Materials and Methods”.
3) Figure 2 and Figure 3 are not clearly labelled. Also the words are blurred.
Reviewer 3 Report
The paper presents the possibility of using two surrogate assays instead of a virus neutralization test to detect neutralizing antibodies in sera against SARS-CoV-2. I have several concerns:
Major comments:
1. The good performances of SARS-CoV-2 surrogate neutralization assays is described and highlighted. They have some advantages such as rapidity, easiness and low cost. However, both surrogate neutralization assays do not depend on physiological conditions, and were not compared with infection blocking/neutralization tests. Due to the limitations of this studies as the authors discussed, there must be potential disadvantages with the surrogate assays. Please make an objective judgment, or at least a full discussion is required.
2. The surrogate assays are based on the interaction/binding of S RBD and ACE2. Although it is a well identified specific process of SARS-CoV-2 infection, other factors are also involved in regulating the infection. For example, antibodies that block the interaction between SARS-CoV-2 and its helper receptor (such as TMPRSS2) also exert neutralization effects. Beside the RBD, the other domain of S protein also contribute to SARS-CoV-2 infection. For example, a cysteine-rich domain (CRD) of S protein contributes to S protein-mediated cell fusion to drive SARS-CoV-2 infection (PMID: 34117209, DOI:10.1038/s41392-021-00651-y). Antibodies that block the activity of S-CRD may also exert neutralization effects. Thus, blockage of the interaction of RBD and ACE2 can be considered as a major means for antibody neutralization, but it is not the only one. Please carry out a more systemic study, or at least a full discussion about these is required.
3. As humans and sera were subjected to the analysis of this study, informed consent forms and an ethical approval for this research were needed. The serial number of ethical approval should be provided in the manuscript so as to ensure the legality of this research.
4. Why were COVID-19 patients vaccinated with 2 doses of vaccine (COVID-19 + 2D) not studied in the trial groups?
Minor comments:
1. Please check the punctuations in Line 85 and Line 151
2. Line 229 “The text continues here.” Is it redundant?
3. It will be better to keep the same orders of the five groups in the description in the text (2.1 Populations tested) and in the list (Table.1)
4. Each image in Figure 2 and Figure 3 should be labeled with the names of SARS-CoV-2 variants. Otherwise, the reader cannot find the corresponding relationship of the data.
5. The legends of the ordinate in Figure 1, “log10 NT100” and “log10 BAU/Ml” are incorrectly written (not a correct label for the values, 1,10,100,1000. If expressed with log10, they should be 0, 1, 2, 3, 4). Please check.
6. Are two vertical lines at the right edge of Fig.3 redundant?
Round 2
Reviewer 3 Report
In the New Version of the manuscript, I cann't see the revisions for the Figures that the authors described in coverletter. Please make sure that they are revised as described.
Author Response
Thanks to the carefull reviewing, we have also modified the figures in the manuscript (the modifications had been added to the figures which were attached separetely to the revised manuscript).